# Pharmacological Effects and Clinical Prospects of Cepharanthine

**DOI:** 10.3390/molecules27248933

**Published:** 2022-12-15

**Authors:** Di Liang, Qi Li, Lina Du, Guifang Dou

**Affiliations:** 1Department of Pharmaceutical Sciences, Beijing Institute of Radiation Medicine, Beijing 100850, China; 2School of Pharmacy, Shandong University of Traditional Chinese Medicine, Jinan 250355, China

**Keywords:** cepharanthine, coronavirus disease 2019, antiviral, novel formulations

## Abstract

Cepharanthine is an active ingredient separated and extracted from *Stephania cepharantha* Hayata, a Menispermaceae plant. As a bisbenzylisoquinoline alkaloid, cepharanthine has various pharmacological properties, including antioxidant, anti-inflammatory, immunomodulatory, antitumoral, and antiviral effects. Following the emergence of coronavirus disease 2019 (COVID-19), cepharanthine has been found to have excellent anti-COVID-19 activity. In this review, the important physicochemical properties and pharmacological effects of cepharanthine, particularly the antiviral effect, are systematically described. Additionally, the molecular mechanisms and novel dosage formulations for the efficient, safe, and convenient delivery of cepharanthine are summarized.

## 1. Introduction

The Menispermaceae family comprises approximately 70 genera with 420 extant species and is considered a medium-sized family. It mainly consists of dioecious climbers and has a variety of medicinal properties [1,2]. The main active ingredients of the Menispermaceae family are alkaloids, which can be classified into six types based on their chemical structure: morphine dienone, lotus alkane, apocynin, proto-apocynin, proto-cotyledonine, and bisbenzylisoquinoline. Cepharanthine is an active ingredient separated and extracted from *Stephania cepharantha* Hayata (Figure 1), which belongs to the bisbenzylisoquinoline type. Cepharanthine has anti-inflammatory, antibacterial, antioxidant, antihemolytic, and immunomodulatory effects [3]. Clinically, it is used to treat snake bites [4], alopecia [5], malaria [6], and radioactive leucopenia [7].

Recently, cepharanthine has become a promising therapeutic molecule owing to its remarkable antiviral effect, particularly the inhibitory effect on severe acute respiratory syndrome coronavirus 2 (SARS-CoV-2), which causes coronavirus disease 2019 (COVID-19). However, its poor water solubility results in low oral bioavailability. Therefore, to improve its solubility and bioavailability, new dosage forms, such as dry powder inhalers (DPIs), liposomes, and nanoparticles, can be prepared, which can be used to provide a more effective dosing regimen for viral infections via pulmonary, oral, and intravenous administration [8].

This review focuses on the important physicochemical and pharmacological properties and molecular mechanisms of cepharanthine and some new dosage forms that can be useful in improving its solubility and bioavailability, providing a basis for its clinical application. Additionally, this review summarizes cepharanthine’s future development prospects.

## 2. Important Physicochemical Properties of Cepharanthine

Cepharanthine is a member of the bisbenzylisoquinoline cyclic alkaloid family [7]. It is also known as 12-O-methyl cepharanoline and is characterized by the presence of a double 1-benzylisoquinoline portion in its alkyl chain. It is mainly found in plants such as *Stephania epigaea* and *S. cepharantha* [9]. Its molecular formula is C_37_H_38_N_2_O_6,_ and it is a generally white or yellow crystalline powder. In the chemical structure of cepharanthine, there are two head-to-head connected coclaurine units, assigning it an elliptical macrocyclic structure. This confers the unique chemical properties of ether solubility, optical activity, and the ability to reduce the mobility of various biofilms [5,7]. Cepharanthine is easily soluble in acidic aqueous solutions and some organic solvents, such as methanol, ethanol, and DMSO, but is hardly soluble in water. It is cationic and amphiphilic [10]. These specific physicochemical properties lead to its low bioavailability. Doses of 1–60 mg per day have been safely and effectively used to treat various conditions. For example, the oral dose for alopecia areata treatment is usually 1.5–6 mg of CEP per day. The dose for the treatment of radiation-induced leukopenia is usually 50–60 mg of CEP per day. The dose for the treatment of multiple myeloma is usually 30 mg of CEP per day [10,11]. CEP has a half-life of 31.3–36.9 h and achieves a steady state after five to six repeated doses of 100 mg/day, with a maximum of 9.6% residual drug after oral administration [12]. The time to reach the maximum serum concentration after a single oral dose of 10–60 mg in healthy men ranges from 1.1 to 2.5 h. After absorption in the body, CEP is extensively metabolized in the liver and distributed to various tissues [5].

## 3. Pharmacological Effects of Cepharanthine

### 3.1. Antiviral Effects

A previous study has indicated that cepharanthine has a potent and extensive antiviral activity against viruses such as SARS-CoV, MERS-CoV, and HIV-1 [13]. It has been demonstrated that in chronically infected U1 cells, cepharanthine inhibits TNF-α or phorbol 12-myristate 13-acetate-induced HIV-1 replication by suppressing NF-κB activation [14]. The anti-HIV-1 activity of cepharanthine is dependent on its ability to inhibit NF-κB, and it inhibits the entry of HIV-1 by reducing plasma membrane fluidity [15]. Additionally, cepharanthine can affect the replication of viruses, such as hepatitis B virus [16], herpes zoster virus [17], and T-lymphotropic virus type 1 [18]. The extensive antiviral activity of cepharanthine may originate from the inhibition of intracellular inflammatory cytokines and chemokines [7].

COVID-19 is caused by a novel coronavirus called severe acute respiratory syndrome coronavirus 2 (SARS-CoV-2), which has a complex mechanism of action. The virus has been reported to affect several vital organs in the body, including the lungs, heart, brain, kidneys, gastrointestinal tract, blood, and immune system [19]. Several vaccines have been developed, but their availability is limited, and they can only have a preventive effect. To date, no specific therapeutic agent for coronavirus infection exists. Several drugs approved for COVID-19 by the US Food and Drug Administration (FDA) have previously been used for other diseases, such as raltegravir (Ebola) [20], lopinavir–ritonavir (AIDS) [21], chloroquine (malaria) [22], and favipiravir (influenza) [23] (Table 1). In early 2020, Yigang et al. discovered that cepharanthine could be a potent drug for treating COVID-19 infection after screening thousands of monomeric compounds [24].

Subsequently, the anti-COVID-19 activity of cepharanthine has been confirmed repeatedly [25,26]. The combined use of nelfinavir and cepharanthine could effectively inhibit SARS-CoV-2 replication, wherein cepharanthine inhibits the entry of SARS-CoV-2 by blocking the binding of the virus to target cells, and nelfinavir inhibits viral replication by inhibiting proteases [27]. Moreover, cepharanthine significantly inhibited the purified recombinant SARS-CoV-2 decapping enzyme, Nsp13, via in vitro enzyme activity assays. Nsp13 is essential for viral replication and is the most conserved nonstructural protein in the coronavirus family [26]. RNA-seq analysis has revealed that cepharanthine may exert antiviral effects through the reversal of heat shock factor-1-mediated heat shock response, endoplasmic reticulum stress/unfolded protein response, and hypoxic pathways of viral interference [28]. A pseudoviral model of SARS-CoV-2 demonstrated that cepharanthine can inhibit SARS-CoV-2 S protein/angiotensin-converting enzyme 2 (ACE2)-mediated membrane fusion by targeting host calcium ion channels and simultaneously upregulating intracellular cholesterol levels, thereby effectively inhibiting infection by SARS-CoV-2 mutants and different coronaviruses [13]. The human coronavirus OC43 model was used to screen 1900 clinically used drugs in vitro, which revealed that cepharanthine had the highest in vitro anti-SARS-CoV-2 activity, much higher than that of the already marketed anti-SARS-CoV-2 drug, raltegravir [29].

The only clinical study to date of the proposed use of cepharanthine in the treatment of COVID-19 is the patent for an enteric formulation of cepharanthine by the Canadian pharmaceutical company, Pharmadrug Inc. This patent prepared cepharanthine as an enteric formulation for oral administration (PD-001), and its bioavailability in animal models was significantly improved. The company has already established GMP manufacturing for PD-001 and FDA-compliant preclinical safety/efficacy studies. Pharmadrug has submitted a Pre-Investigational New Drug Application to the FDA for the treatment of mild-to-moderate COVID-19 and will be working with the regulator to bring this medication for a complete evaluation.

### 3.2. Prevention of Leukopenia

Cepharanthine was first used to increase the number of leukocytes in the peripheral blood of patients undergoing radiotherapy or chemotherapy [7]. The major mechanisms involved are now widely recognized as the stimulation of the reticuloendothelial system, the activation of hematopoietic tissue, and the promotion of bone marrow proliferation, which ultimately increases the white blood cell count [30].

Leukopenia is clinically defined as a reduction in the number of neutrophils and can be caused by many factors, including chemotherapy and radiotherapy for tumors and chemicals such as benzene. In the 1930s and 1940s, cepharanthine was used to treat pulmonary tuberculosis. It was found that patients with tuberculosis had an increased number of leukocytes in the peripheral blood after cepharanthine administration, which indicated the prospect of cepharanthine in preventing leukopenia. In several clinical studies involving more than 350 patients, radiotherapy was used along with cepharanthine to treat head and neck tumors and ovarian, lung [10], and breast cancers [31]. All studies showed that cepharanthine helped prevent leukopenia in patients receiving anticancer treatment.

### 3.3. Antitumor Effects

Cepharanthine exhibits antitumor effects by enhancing immunity [32] and inhibiting tumor cell proliferation [33], as well as increasing the sensitivity of tumor cells to radiotherapy while reducing the adverse effects caused by the radiotherapy [34]. A more promising approach is the use of cepharanthine in combination with other chemotherapeutic drugs to reverse multidrug resistance (MDR) of tumor cells [35]. CEP effectively inhibits the drug transporter protein ABCC10 (also known as MRP7) on cancer cell membranes, thereby reversing anticancer drug resistance in cells expressing ABCC10. This allows for the increased accumulation of anticancer drugs in cancer cells and inhibits their efflux [36], thus improving the efficacy of anticancer drugs. CEP also interferes with other transporter proteins, such as ABCB1 (also known as MDR1 or P-glycoprotein) [37], possibly by inhibiting the PI3K/Akt signaling pathway, leading to the downregulation of ABCB1 expression in cancer cells, which in turn reverses MDR [38]. CEP potentially interacts directly with specific drug transporter proteins present on the plasma membrane. For example, neferine, another bisbenzylisoquinoline alkaloid with a similar structure to CEP, binds strongly to and inhibits ABCB1, thereby reversing MDR. Therefore, it is speculated that CEP has a similar biological effect [39].

Cepharanthine can be involved in tumor prevention and treatment through various mechanisms, including increasing the white blood cell count and exerting immunomodulatory effects on macrophages, T cells, and natural killer cells to increase the immune capacity [32,40]. Additionally, it can play a role in inducing apoptosis, inhibiting tumor cell infiltration and metastasis.

### 3.4. Anti-Inflammatory Effects

Cepharanthine is also effective for inflammation in vivo, especially in the ears, nose, throat, and mouth. It can significantly decrease the expression of certain inflammatory cytokines, such as IL-6, IL-1β, and TNF-α [41]. In a mouse model of lipopolysaccharide (LPS)-induced mastitis, cepharanthine significantly reduced neutrophil infiltration and decreased TNF-α, IL-1β, and IL-6 levels [41]. Presently, cepharanthine is used to treat inflammatory diseases such as rheumatism, arthritis, lumbago, nephritis, edema, and dysentery [7].

### 3.5. Immunomodulation

Cepharanthine can be used as an immunomodulator, and it has excellent potential in the treatment of various autoimmune diseases and allergies [42]. At a low dose, cepharanthine could effectively prevent progressive thrombocytopenia and was used to successfully treat a patient with multiple myeloma combined with immune thrombocytopenic purpura [43]. Additionally, cepharanthine specifically inhibits abnormally activated T cells or steroid-resistant human leukemia T cells [44,45]. Furthermore, cepharanthine regulates several signaling pathways in abnormally activated T cells, such as NF-κB, caspase cascade, cell cycle, mitogen-activated protein kinase (MAPK), and PI3K/Akt/mTOR pathways [46].

## 4. Molecular Mechanisms of Cepharanthine

### 4.1. NF-κB Pathway

The NF-κB pathway plays a central role in inducing pro-inflammatory gene expression and is a potent regulator of inflammatory molecules. In resting cells, NF-κB exists in the cytoplasm as a latent, inactive IκB-binding complex. Upon relevant stimulation, IκB kinase (IKK) induces phosphorylation of the IκB protein via the ubiquitin–proteasome pathway, resulting in its rapid degradation, allowing NF-κB dimers to enter the nucleus and activating specific target gene expression [46]. In an LPS-stimulated model of RAW264.7 cells, cepharanthine inhibited NF-κB activation by blocking the IKK pathway [47] (Figure 2). In this study, cepharanthine (2.5, 5, and 10 μg/mL) inhibited the release of pro-inflammatory factors such as TNFα, IL-6, and IL-1β in a dose-dependent manner with potentially no cytotoxicity. In addition, 10 mg/kg/day of cepharanthine could significantly inhibit the expression of NF-κB and reduce the levels of the pro-inflammatory cytokines, IL-1β and TNF-α, in diabetic rats [48]. Thus, it seems that cepharanthine can reduce the expression of inflammatory factors in related disease models through the NF-κB pathway and exert anti-inflammatory effects as a potential drug for the treatment of certain related diseases.

### 4.2. Apoptosis

Apoptosis is a regulated cellular suicide mechanism characterized by nuclear condensation, cell shrinkage, membrane leakage, and DNA fragmentation [49]. The cascade of caspases, which belong to the cysteine protease family, is a key apoptosis regulator. To date, only two types of caspases have been defined, including initiator and effector caspases. Initiator caspases, such as caspase-2, -4, -8, -9, -10, and -12, are closely associated with proapoptotic signaling. Upon stimulation by apoptosis, these initiator caspases divide and activate downstream effector caspases, such as caspase-3, -6, and -7. These effector caspases hydrolyze a range of substrates, generating signals that lead to cell death [50,51]. The cell growth inhibition of primary effusion lymphoma (PEL) cell lines was enhanced in a dose-dependent manner as the cepharanthine dose was increased from 1 to 10 μg/mL. Moreover, cepharanthine-treated PEL cells induced the activation of caspase-3, which is an apoptosis marker [52]. In addition, cepharanthine at high concentrations (10–20 μg/mL) decreased the mitochondrial membrane potential of dendritic cells, and high levels of cellular stress caused the release of cytochrome c from mitochondria. This activates caspase-9, which then activates the effector caspase-3/7, and finally triggers apoptosis [53].

### 4.3. Cell Cycle Control

The cell cycle is regulated by different cellular proteins, such as the cell cycle protein A/B/D [54]. It has been shown that cepharanthine affects the cell cycle, usually arresting cells in the G1 and S phases. Jurkat T cells treated with 5, 10, and 15 μΜ CEP showed a dose-dependent inhibition of cell cycle progression in the S phase, significantly reducing the number of cells in the G0/G1 phase [45]. Further research has revealed that cepharanthine upregulates the expression of cell cycle proteins A2 and B1 but downregulates that of the cell cycle protein D1 in Jurkat T cells, possibly relating to cell cycle arrest [46]. Using 0, 1, 5, 10, and 20 μM cepharanthine also effectively reverses MDR-mediated resistance to cisplatin in esophageal squamous cell carcinoma (ESCC) cells, inhibiting the proliferation of ESCC cell lines and inducing G2/M phase cell cycle arrest. Dose-dependent upregulation of p53 and downstream p21 induced by cepharanthine explains its mechanism of causing cell cycle arrest [55].

### 4.4. MAPK Pathway

Activated MAPK has been reported to play an important role in promoting and maintaining T lymphocyte populations [56]. Cepharanthine regulates multiple signaling pathways that aberrantly activate T cells at low toxicity, and the antiproliferative effect on human peripheral blood T cells occurs partly through the blockade of MAPKs, such as JNK, p38, and ERK activity [57]. It was reported that another class of bisbenzylisoquinoline alkaloids, tetrandrine, in combination with methylprednisolone (a glucocorticoid), significantly inhibited the phosphorylation of the mitogen-activated protein kinase family. Powdered tetrandrine (3 μM) in combination with 0.5 ng/mL methylprednisolone showed synergistic inhibition of both ERK1/2 and P38. The powdered antifungal base significantly reduced the IC50 value of methylprednisolone but had no significant toxic effect on normal cells [46]. These evaluations suggest that CEP, a member of the bisbenzylisoquinoline alkaloid family, may have similar efficacy and could be used as a lead compound for the development of new drugs for the treatment of T-cell-related diseases or to address glucocorticoid resistance.

### 4.5. PI3K/Akt/mTOR Signaling Pathway

The PI3K/Akt/mTOR signaling pathway is involved in regulating various cellular responses, such as metabolic regulation [58], cell proliferation, transcription, translation, survival, and growth [58,59]. This pathway is essential for the pathological and physiological conditioning of humans, and changes in the regulation of this pathway may lead to the development of various cancers [60]. Thus, the PI3K/Akt/mTOR signaling pathway is a potential target for antitumor therapy. In a study on breast cancer cells, treatment with 5 and 10 μM cepharanthine was found to reduce the levels of both phosphorylated AKT and mTOR. The downstream targets of mTOR (p70S6K, p-S6, and p-4E-BP1) were significantly reduced, whereas GSK3β, a downstream effector of AKT [61], showed similarly reduced levels in cells after CEP treatment. These results suggest that cepharanthine induces apoptosis and autophagy in breast cancer cells by inhibiting the AKT/mTOR signaling pathway[62]. Cepharanthine can inhibit the expression of p-PI3K and mTOR in Jurkat T cells; however, this resulted in high expression of p-Akt1 [45]. This result may appear to be incongruous, as Akt is considered a major downstream effector of PI3K in physiological processes [63]. However, several studies have demonstrated that different groups of tyrosine (Ack1/TNK2, Src, and PTK6) and serine/threonine (TBK1, IKBKE, and DNAPKcs) kinases can directly activate Akt [64]. Thus, CEP may regulate p-PI3K and mTOR expression independently of Akt.

### 4.6. P-glycoprotein Expression

P-glycoprotein (also known as ABCB1) belongs to the ABC superfamily of transporter proteins [65] and is encoded by multidrug resistance gene 1 (*MDR-1*), which is an ATP-dependent membrane transporter protein [66]. The overexpression of drug transporter proteins present in cancer cell membranes is a major cause of MDR to cancer chemotherapy. Cepharanthine can reverse the resistance of tumor cells to many chemotherapy drugs by interfering with P-glycoprotein [37]. Cepharanthine may act as a potent P-glycoprotein inhibitor, dose-dependently restoring antitumor activity [67]. In a study on glucocorticoid resistance, the human T-lymphoblast leukemia cell line MOLT-4 with low P-glycoprotein expression, and its MDR subline MOLT-4/DNR, were used as the model. Tetrandrine (Figure 3), being likewise a bisbenzylisoquinoline-like alkaloid and highly resembling cepharanthine, indirectly modulated the translocation of glucocorticoid receptors by inhibiting the efflux function of P-glycoprotein in MOLT-4/DNR cells and downregulated its protein expression, thereby enhancing the pharmacodynamic effects of glucocorticoids. The expression of P-glycoprotein was downregulated in MOLT-4/DNR cells in a concentration-dependent manner under treatment with 0.03, 0.3, and 1 μM tetrandrine. Although 1 μM tetrandrine significantly inhibited P-glycoprotein expression, MOL T-4/DNR 296 cells exhibited cytotoxic effects at this dose. In contrast, glucocorticoid receptor translocation was almost completely restored at 0.3 μM with no notable side effects [68]. However, studies on the resistance of CEP to glucocorticoids remain unclear, and the actual effective dose needs to be explored further. However, it can be deduced that this bisbenzylisoquinoline alkaloid of natural plant origin is beneficial for glucocorticoid-resistant diseases caused by P-glycoprotein overexpression.

## 5. New Dosage Forms of Cepharanthine

Presently, the commercial formulations of cepharanthine are ordinary tablets. As cepharanthine is insoluble in water, it is distributed in tablets in the form of larger granules. It is administered at larger doses, and the rate of dissolution and diffusion after oral administration is slow, significantly affecting its absorption and utilization in vivo. Therefore, it is important to explore new dosage forms of cepharanthine to widen its clinical application.

### 5.1. Oral Formulation

#### 5.1.1. Oral Disintegrating Tablets

Oral administration is the most widely used and convenient route of administration, with a high degree of stability and portable packaging [69]. Orally disintegrating tablets, as a delivery system, are those that disintegrate or dissolve rapidly in the mouth without needing water and are absorbed through the antral mucosa [70]. Compared with ordinary tablets, orally disintegrating tablets have a faster onset of action, avoid the liver first-pass effect, have higher bioavailability, are less irritating to the esophagus and gastrointestinal tract, and have higher compliance. Therefore, they are ideal for children, elderly patients, and patients with swallowing difficulties [71,72]. The preparation methods include conventional techniques such as freeze drying [73], spray drying [74], direct compression [75], and molding [76] and new techniques such as marshmallow technology [77], microwave-assisted irradiation [78], sublimation [79], and 3D printing [80]. It is possible to consider formulating cepharanthine as an orally disintegrating tablet to improve its bioavailability and reduce its dose.

#### 5.1.2. Dropping Pills

As a form of solid dispersion, dropping pills are the preparations made by heating or mixing drugs with a matrix, then dropping them into an immiscible condensate, causing the droplets to shrink and condense due to surface tension [81]. Unlike conventional tablets, dropping pills have the advantages of small dosage, fewer adverse effects, fast dissolution, high bioavailability, good efficacy, easy administration, excellent drug stability, simple preparation, and easy quality control [82]. They are good candidates to improve drug dissolution and increase oral bioavailability. In the preparation of dropping pills, a solid dispersion is formed, and drugs can be present in the form of separated molecules or amorphous particles, which substantially improves their solubility [83]. Cepharanthine dropping pills have been prepared with PEG 4000 and PEG 6000 serving as the excipients using the solid dispersion technology to improve their solubility, thereby achieving rapid and high efficiency [84].

### 5.2. Injections

Although oral administration is the preferred route of administration due to its convenience and noninvasiveness, it is impossible to deliver all drugs orally due to low bioavailability and patients’ tolerance to oral intake. Therefore, injections can be an effective alternative [85]. The advantages of injections include rapid efficacy, direct entry into the bloodstream when administered without passing through the gastrointestinal tract, improved bioavailability and reduced drug interactions, targeted efficacy, and suitability for drugs that are unsuitable in oral dosage forms or for patients who cannot receive oral administration [86].

After the COVID-19 outbreak, several Chinese medicine injections, including the Reduning injection [87], Xiyanping injection [88], and Xuebijing injection [89], were used as specific therapeutical agents. The Chinese pharmaceutical company BUCHANG has applied for a patent on the preparation and testing method of cepharanthine hydrochloride injection. The preparation method is simple and controllable, with reliable quality and high bioavailability, which can overcome many disadvantages of oral administration of cepharanthine and has good application prospects.

### 5.3. Pulmonary Drug Delivery Systems—DPIs

A pulmonary drug delivery system is a system that delivers drugs to the lungs to produce local or systemic therapeutic effects [90]. Compared with conventional drug delivery, pulmonary drug delivery acts faster, avoids hepatic first-pass effects, reduces the dose required and adverse effects associated with systemic drug delivery, and substantially improves patient compliance [91]. Currently, there are three types of pulmonary delivery systems: metered-dose inhalers, DPIs, and nebulizers. DPI sales account for approximately 50% of all global sales of pulmonary drug delivery systems, far exceeding the sales of the other two dosage forms. It is safe and environmentally friendly as no projectile agent or preservative is used in the DPI. Furthermore, the stability of the drugs existing as a dry powder is significantly improved [92]. Common processes for preparing DPIs include spray drying, freeze drying, airflow pulverization, supercritical fluid pulverization, grinding and pulverization, etc. [93]. The ideal DPI should meet the following criteria: high dispersibility, drug stability, narrow aerodynamic particle size distribution, low particle surface energy and potential, good dose reproducibility, and a sustained release or specific targeting mechanism when required [94].

Cepharanthine is a water-insoluble drug that is appropriate as a DPI for pulmonary inhalation to treat COVID-19 associated with its sequelae and complications because the vital infection organ of SARS-CoV-2 is the lung.

### 5.4. Nanoformulations

#### 5.4.1. Liposomes

Liposomes are miniature vesicles formed by encapsulating a drug within a lipid-like bilayer. It is considered a highly promising drug delivery system because of its biofilm-like structure, ability to encapsulate both water and fat-soluble drugs, ability to reduce drug dose, reduce toxicity, mitigate metabolic and immune reactions, delay release, reduce the rate of elimination in the body, alter drug distribution in the body, and its ability to target drug release [95]. The liposome encapsulation technology can solve the challenge of drug dissolution. By encapsulating unstable, oxidizable drugs in liposomes, these drugs are protected by lipid membranes, which can reduce the probability of events such as enzymatic degradation, chemical and immunological inactivation, and rapid plasma clearance, helping improve and prolong their action [96]. Among the hundreds of nanoformulations approved by the FDA, liposomes account for the highest percentage (33%) [97]. Furthermore, various administration routes, such as injection, oral, transdermal, transnasal, transpulmonary, and ophthalmic, have been developed.

As a water-insoluble drug, cepharanthine can be designed as a formulation encapsulated in liposomes for transpulmonary or oral administration for treating COVID-19. This solves the problem of the low solubility of cepharanthine in water and improves its bioavailability in vivo.

#### 5.4.2. Nanoparticles

Nanoparticles are sphere-like particles formed by dissolving or encapsulating a drug in a polymeric material, usually between 10 nm and 100 nm in size, which are more effectively absorbed by cells than larger molecules [98]. In recent years, nanoparticles have been widely used for drug delivery and have proven to be effective in drug delivery and diagnostics [92]. Compared with other types of drug-carrying particles, nanoparticles have a higher drug-carrying capacity [99], permeate permeability barriers, prolong circulation times [100], and increase cellular uptake [101], thus improving safety and efficacy. Nanoparticles typically have a large surface-area-to-volume ratio, and this feature improves their dissolution properties, thereby increasing solubility and intracellular drug delivery potential. However, engineered nanoparticles are foreign to the organism, and once they enter the body, they immediately activate the innate immune system and generate a specific immune response based on their properties [102]. Currently, it is possible to evade the immune response by designing nanoparticles that mimic host cells [103], adjusting their physicochemical properties such as preparing them in different sizes and shapes with a surface charge [104], or modifying the surface of the particles with a bionic coating [105].

Bionic anti-inflammatory nanoformulations of cepharanthine for the treatment of acute lung injury (ALI) have been reported, which are based on the “homing” of macrophages to sites of inflammation and cell membrane coating nanotechnology. Researchers used macrophage membranes isolated from the RAW264.7 monocyte/macrophage cell line to encapsulate nanostructured lipid carriers loaded with cepharanthine. The nanoparticles were then injected into the lung inflammation sites of ALI mice via tail vein injection. Long-term circulation was achieved in vivo and showed the ability to target inflamed lungs [106].

### 5.5. Summary

Based on the above-listed several new dosage forms of cepharanthine, we have summarized their advantages and disadvantages as well as those that need attention during clinical development. Orally disintegrating tablets are convenient to administer and highly stable, while avoiding the first-pass effect in the liver and improving the low bioavailability of cepharanthine. However, orally disintegrating tablets have the disadvantages of fragility and moisture absorption; therefore, there is still a need to improve the formulation and packaging materials to enhance their application. Another oral formulation, drops, in the form of a solid dispersion formed during its preparation can effectively improve the solubility of cepharanthine while enhancing its bioavailability. However, quality control of this new dosage form is difficult, and quality standards are lacking; therefore, there are no new dosage forms of drops approved for marketing. Injections are another drug delivery method that have rapid efficacy and are suitable for patients who cannot receive oral administration. However, injectable drug delivery faces the challenge of poor water solubility of active ingredients, and suitable solubilization strategies need to be adopted according to the chemical structure and physicochemical properties of insoluble drugs. In addition, pulmonary drug delivery systems are superior to traditional drug delivery methods for their advantages, such as fast-acting effects and avoidance of the hepatic first-pass effect. Cepharanthine is a water-insoluble drug suitable for use as a DPI for pulmonary inhalation. What requires attention when developing dosage forms is designing the formulation so that it meets the ideal DPI criteria. Nanoformulations, including liposomes and nanoparticles, have been widely used for drug delivery in recent years as a new dosage form. They can encapsulate water-insoluble drugs in microcapsules or particles to achieve targeted and delayed drug release. However, these substances are foreign to the organism and can stimulate an immune response when entering the body. Therefore, the most important factor when designing a nanoformulation is addressing the immune rejection reaction.

## 6. Conclusions

Cepharanthine has become a promising molecule due to its excellent antiviral activity against SARS-CoV-2. This review summarizes the important physicochemical properties and pharmacological effects of cepharanthine (Table 2), including its anti-COVID-19 effect. Additionally, the molecular mechanisms (Table 3) and novel dosage formulations for the efficient, safe, and convenient delivery of cepharanthine are summarized, including oral, injectable, DPIs, and nanoformulations. The number of studies on cepharanthine has shown exponential growth following the discovery of the remarkable potential of cepharanthine, an “old” drug derived from a natural plant, in the fight against COVID-19. However, cepharanthine is mainly used clinically for treating leukopenia caused by radiotherapy and chemotherapy for tumors, while its excellent anti-COVID-19 activity has only been demonstrated in vitro and at the cellular level. Therefore, it is important to determine methods to improve the bioavailability of cepharanthine in vivo so that it could exert a marked anti-SARS-CoV-2 effect in vivo, making it truly applicable in COVID-19 treatment.

## Figures and Tables

**Figure 1 molecules-27-08933-f001:**
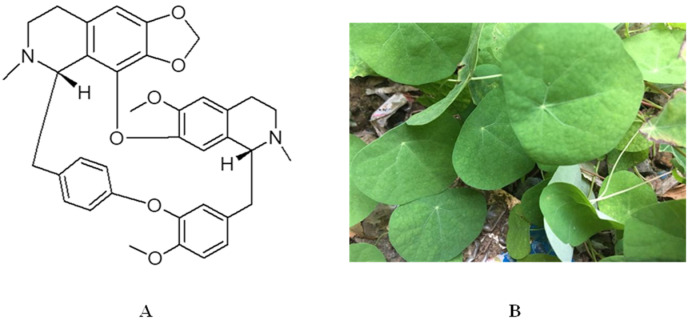
(**A**) Chemical structure of cepharanthine and (**B**) plant image of *Stephania cepharantha* Hayata.

**Figure 2 molecules-27-08933-f002:**
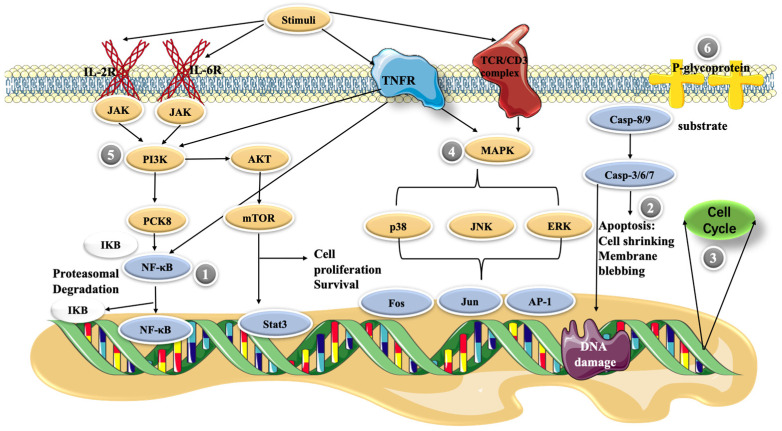
Possible molecular mechanisms of cepharanthine in cells. (**1**) NF-κB pathway, (**2**) apoptosis, (**3**) cell cycle control, (**4**) MAPK pathway, (**5**) PI3K/Akt/mTOR signaling pathway, and (**6**) P-glycoprotein expression.

**Figure 3 molecules-27-08933-f003:**
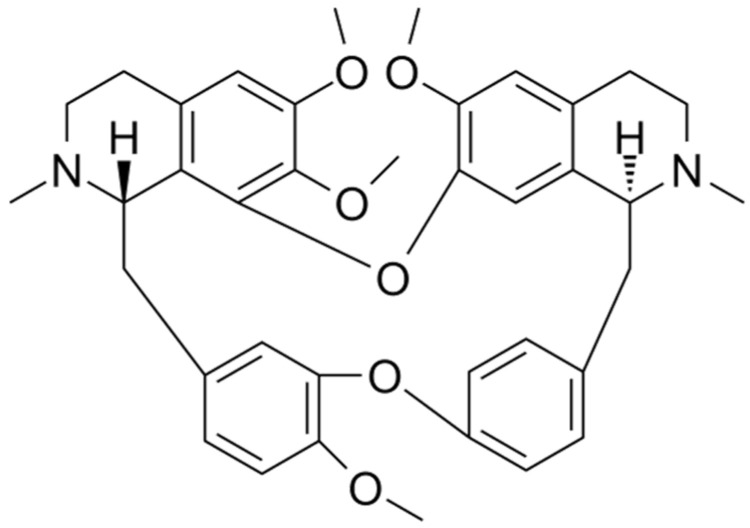
Chemical structure of tetrandrine.

**Table 1 molecules-27-08933-t001:** Approved drugs with anti-COVID-19 viral activity.

Drugs	Conventional Use	Dosage	Administration Routes
Interferon-α	Hepatitis, pneumonia	5 million U (2 mL of sterile water for injection) each time, 2 times/day	Vapor inhalation, intramuscular injection
Lopinavir–ritonavir	Human immunodeficiency virus	400 mg/100 mg each time, 2 times/day	Oral
Ribavirin	Viral infection	500 mg each time, 2–3 times/day in combination with IFN-α or lopinavir/ritonavir	Intravenous infusion
Chloroquine phosphate	Malaria	500 mg (300 mg for chloroquine) each time, 2 times/day	Oral
Arbidol	Infection of the upper respiratory tract	200 mg each time, 3 times/day	Oral
Favipiravir	Flu virus	On day 1, 800 mg each time; on days 2–5, 300 mg each time, 2 times/day	Oral
Remdesivir	Respiratory virus, hepatitis C	On day 1, 200 mg/day; on days 2–9, 100 mg/day	Intravenous infusion

**Table 2 molecules-27-08933-t002:** Pharmacological effects of cepharanthine.

Property	Mechanism of Action	References
Antiviral effects	HIV-1: Reduces plasma membrane fluidity	[15]
	HBV, HZV, HTLV-1: Affects virus replication	[16,17,18]
	SARS-CoV-2: Blocks the binding of the virus to target cells, inhibits the purified recombinant SARS-CoV-2 decapping enzyme Nsp13, inhibits SARS-CoV-2 S protein/angiotensin converting enzyme2 (ACE2)-mediated membrane fusion by targeting host calcium ion channels, and upregulates intracellular cholesterol levels	[13,26,27]
Prevention of leukopenia	Stimulates the reticuloendothelial system, activates hematopoietic tissue, and promotes bone marrow proliferation	[30]
Antitumor effects	Enhances immunity, inhibits tumor cell proliferation, increases tumor cell sensitivity to radiotherapy, inhibits tumor cell infiltration and metastasis, and reverses multidrug resistance of tumor cells	[32,33,35,52]
Anti-inflammatory effects	Reduces neutrophil infiltration and decreases TNF-α, IL-1β, and IL-6 levels	[41]
Immunomodulation	Prevents progressive thrombocytopenia and regulates several signaling pathways in abnormally activated T cells	[43,46]

**Table 3 molecules-27-08933-t003:** Detailed molecular mechanisms of cepharanthine in cells.

Targets	Detailed Molecular Mechanisms
NF-κB	Inhibits NF-κB activation by blocking the IKK pathway in RAW264.7 cells. Inhibits the phosphorylation of the NF-κB p65 subunit and the degradation of its inhibitor IκBα in lipopolysaccharide-induced mastitis.
Apoptosis	Activates caspase-9, then activates caspase-3/7 and triggers apoptosis. Upregulates the expression of caspase-8/9 and caspase-3/6 in glucocorticoid-resistant Jurkat T cells in vitro.
Cell cycle control	Dose-dependently inhibits the cell cycle progression of Jurkat T cells in the S-phase. Upregulates the expression of cell cycle proteins A2 and B1 but downregulates that of the cell cycle protein D1 in Jurkat T cells. Dose-dependently upregulates p53 and downstream p21.
MAPK	Activates p38, slightly stimulates JNK phosphorylation, and potentially induced apoptosis. Activates oxidative stress and stress-activated kinases JNK1/2, MAPK p38, and ERK and induces chromatin condensation and nuclear fragmentation.
PI3K/Akt/mTOR	Induces apoptosis and autophagy by inhibiting the AKT/mTOR signaling pathway in breast cancer cells. Inhibits the expression of p-PI3K and mTOR on AKT in Jurkat T cells.
P-glycoprotein	Activates the JNK signaling pathway and induces the expression of MDR1 mRNA and P-glycoprotein. Inhibits the efflux function of P-glycoprotein in MOLT-4/DNR cells and downregulates its protein expression.

## Data Availability

Not applicable.

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
