# Peer review of "Pharmacological Effects and Clinical Prospects of Cepharanthine"

_molecules, 2022, doi:10.3390/molecules27248933_

Round 1

Reviewer 1 Report

Molecules-2059628

The manuscript entitled “Pharmacological effects and clinical prospects of cepharanthine” presents an overview of pharmacological properties of cepharanthine and possible novel dosage formulations for its efficient and safe administration in patients.

The review is well written and the paragraphs cover several features of the subject. The exposition is clear and interesting.

However, I have some suggestions.

In the Figure 1 the letters A and B don’t indicate the correct image. Please correct.

In the legend of the Figure 1 it would be better to use the term “plant image” instead of “plant morphology”.

In the Introduction section a brief description of Menispermaceae family should be added before introducing  the cepharanthine.

Paragraph 2 should be improved by introducing more information on the chemical structure that confers chemical properties to cepharanthine together with the fate when administered to patients and the biodistribution and metabolism prior to excretion.

Regarding the  pharmacological effects of cepharanthine (such as anti-Sars CoV2 effect, prevention of leukopenia etc.), a table including studies and related references should be added after Table 1.

The molecular mechanisms of cepharanthine are illustrated in Figure 2. Table 2 reports the same information found in Figure 2. Table 2 should be deleted or alternatively Figure 2 should be deleted.

Regarding paragraph 5, it is better to first describe the new dosage forms for which cepharanthine studies already exist, such as dropping pills, followed by the dosage forms which appear promising for cepharanthine clinical application.

The Conclusion section should be improved by summarizing the advantages of the new proposed dosage formulations together with the potential risks associated with the components and methods used to the development that could affect their clinical applications.

Reviewer 2 Report

In their review on cepharantine, Di Liang et al. mainly describe the various pharmacological effects of this compound and the various dosage formulations which can be used. The latter are important to aim for improved uptake in view of the low solubility of the lipophilic compound, and to enable more detailed clinical evaluations.

Overall, this review is well written but in general only summarizes the long list of literature findings without much discussion or expression of an expert opinion on the sometimes conflicting or unclear data. Only a personal evaluation and opinion writing about the various aspects makes such reviews really worthwhile. I therefore would urge to add such reflective paragraphs throughout the manuscript.

In contrast, the formulation part on new dosage forms for cepharanthine is interesting and these aspects are less known, and serve the scientific community in order to have this product more readily available for further clinical evaluation studies.

The introduction states that the compound is non-toxic (line 53-54: Cepharanthin can be administered orally or intravenously at high doses for a long time with minimal or no adverse effects [5,9]). Yet the compound is claimed to interfere in numerous pathways. How can this be explained? I presume a lot of these pharmacological effects are only seen at high and possibly anecdotal doses.

Along the same lines, the manuscript is heavily documented by citations, but are all these results scientifically relevant?  Could the authors therefore discuss the required doses which are needed according to the cited literature for inhibition of each of the various pathways? Please elaborate.

Interaction in several pathways is claimed in literature, but which ones really have been proved via inhibition of a specific enzyme, and what were the binding affinities?

Further remarks:

Figure 1 – line 29: the panels A and B are reversed

Line 45: the characteristic part of the molecule is present twice. Therefore, adapt to “is characterized by the double presence of a 1-benzylisoquinoline portion in its alkyl chain.”

Lines 68-83: the long elaboration on covid-19 seems a bit superfluous in the context of the pharmacological effects of cepharanthine and this paragraph can be shortened.

Lines 130-133 are a slightly rephrased repetition of lines 117-120 and therefore can be removed (or at least formulated differently).

Lines 147-148:  is it known how exactly cepharantine is modulating the plasma membrane to enable increased uptake of various products? Such discussions would be at the benefit of this review.

Line 148: reference 37 and 38 are used to sustain the above claim, but these references apparently have no link whatsoever with cepharanthine. This term is actually not found in the cited papers. Please add the corrected citations.

Line 332: rephrase this sentence as the challenge is not to make lipid-soluble drugs, but to enable dissolution of these drugs.

Line 376: insert the word “which” (to read “which were analyzed and compared”)

Line 377:  correct to “The number of studies on cepharanthine have shown exponential growth….” 

Round 2

Reviewer 2 Report

In the revised version of their manuscript on cepharantine, Di Liang et al. carefully updated their review taking into account the remarks of both referees. Several paragraphs were added to further highlight the pharmacological interactions taking into account the required non-toxic doses. Further discussion on the various formulations likewise was added. The manuscript this way considerably improved and provided some small changes mentioned here below, can be accepted.

Line 268: I would situate the “tetrandrine” a bit more as the structure itself is not shown in the manuscript. Suggested phrasing: “Tetrandrine, being likewise a bisbenzylisoquinoline-like alkaloid, and highly resembling to cepharanthine, ….”

Line 278: correct to “bisbenzylisoquinoline” (from bibenzyl)

Figure 2: in agreement with the authors’ reply, I likewise support to keep this figure in the manuscript.

Line 412: what is seneciostin? I failed to google it and why does this end up in the conclusion section (which is discussing solubility of cepharantin)

Lines 406-429: This part on the advantages and shortcomings of the new dosage forms, which was added to the conclusion, preferably should be added as a section 5.5 as part of the discussion, and not in the conclusion section. In addition, possibly some references could be added for this specific part.
